# Long-Term Clinical Outcomes of Treatment with Dental Implants with Acid Etched Surface

**DOI:** 10.3390/ma13071553

**Published:** 2020-03-27

**Authors:** Eugenio Velasco-Ortega, Alvaro Jimenez-Guerra, Loreto Monsalve-Guil, Ivan Ortiz-Garcia, Ana I. Nicolas-Silvente, Juan J. Segura-Egea, Jose Lopez-Lopez

**Affiliations:** 1Comprehensive Dentistry for Adults and Gerodontology, Faculty of Dentistry, University of Seville, Calle San Fernando, 4, 41004 Sevilla, Spain; evelasco@us.es (E.V.-O.); alopajanosas@hotmail.com (A.J.-G.); lomonsalve@hotmail.es (L.M.-G.); ivanortizgarcia1000@hotmail.com (I.O.-G.); 2School of Dentistry, University of Murcia, Marques de los Velez, s/n, 30008 Murcia, Spain; 3Faculty of Dentistry, University of Seville, Calle San Fernando, 4, 41004 Sevilla, Spain; segurajj@us.es; 4Faculty of Dentistry, Medical-Surgical Area of Dentistry Hospital, University of Barcelona, Gran Via de les Corts Catalanes, 585, 08007 Barcelona, Spain; jl.lopez@ub.edu

**Keywords:** implant dentistry, acid-etched surface, early loading, osseointegration, long-term evaluation

## Abstract

Implant dentistry constitutes a therapeutic modality in the prosthodontic treatment of partially and totally edentulous patients. This study reports a long-term evaluation of treatment by the early loading of acid-etched surface implants. Forty-eight partially and totally edentulous patients were treated with 169 TSA Defcon® acid-etched surface implants for prosthodontic rehabilitation. Implants were loaded after a healing free-loading period of 6–8 weeks in mandible and maxilla, respectively. Implant and prosthodontic clinical findings were followed during at least 17 years. Clinical results indicate a survival and success rate of implants of 92.9%, demonstrating that acid-etched surface achieves and maintains successful osseointegration. Five implants in three patients were lost during the healing period. Sixty-five prostheses were placed in 45 patients over the remaining 164 implants, 30 single crowns, 21 partially fixed bridges, 9 overdentures, and 5 full-arch fixed rehabilitations. A total of 12 implants were lost during the follow-up period. Mean marginal bone loss was 1.91 ± 1.24 mm, ranging from 1.1 to 3.6 mm. The most frequent complication was prosthetic technical complications (14.2%), followed by peri-implantitis (10.6%). The mean follow-up was of 214.4 months (208–228 months). Prosthodontic rehabilitation with an early-loading protocol over acid-etched surface implants is a successful implant treatment.

## 1. Introduction

Implant treatment of patients with partial or total tooth loss constitutes a well-documented surgical and prosthodontic therapy [1,2]. The surface of dental implants plays an essential role in osseointegration. Osseointegration is related to the interaction between the implant surface and the host´s bone tissue [3]. The surface treatment might modify the roughness, the topography, and the surface chemical composition, and lead to different biological responses [4,5]. The surface treatment can also increase the contact area between the implant and the hosting bone and improve both the cell migration and the attachment to the implant, enhancing the osseointegration process [6,7]. There is a need to modify the surface to improve the interaction with the extracellular environment and to enhance osteogenic responses as cell proliferation, adherence, and differentiation [6,7].

After the introduction of machined implants, surface modifications have been developed and applied on implants by different additive techniques (such as titanium plasma spray or coating), by subtractive techniques (such as sandblasting or acid etching) or by a combination of both additive and subtractive techniques, being the combination of sand-blasting and acid-etched the most widely used [8,9]. Sandblasting is a common procedure used for the treatment of the implant surface performed by a projection of alumina, hydroxyapatite, silica, or TiO_2_ particles_._ The effect achieved on the surface implant depends on the type, particle size, pressure, and temperature [10,11].

Acid-etching is a conventional treatment for dental implant surfaces. In this technique, dental implants are immersed in acidic solutions [12,13] with a combination of different acids (such as sulphuric, nitric, or hydrofluoric). The resulting surface features after the procedure are related to the type and concentration of the acids, the time of exposition, and the temperature. The surface presents erosion with the formation of peaks and cavities with different highs and wides [14]. Titanium samples etched by acids with different concentrations demonstrated an increase in the surface roughness for biological applications [15].

A relevant osteogenic response can be improved by implant surface treatment with acid-etching [7,14]. A recent in vitro study demonstrated that treating the implant surface by acid-etching plays a vital role in osseointegration. The study assesses different implant surfaces (i.e., machined, titanium plasma spray, sandblasted, acid-etched). The acid-etched surface showed a positive influence on the proliferation, adherence, and differentiation of osteogenic cells. Actually, the acid-etched surface had the highest cell proliferation rate of implant surface studies [6].

There are other surface treatments, such as electrochemical anodization of Ti, which combines the titanium etching with the oxide growth when a voltage is applied, resulting in the formation of porous titania nanotubes that grows perpendicular to the metal (anodic porous titania). The presence of the nanoporous oxides provides biocompatibility and tunes the roughness for an optimized stimulation of living-cell response, even on curved surfaces, leading to an improvement for the nanopatterning of dental implant surfaces. The anodization can be an additive treatment in addition to other surface treatments, such as sandblasting or acid-etching [16].

The implant removal by reverse torque constitutes an indirect method for measuring the osseointegration because it provides information about the required force to fracture the bone-implant interface [17]. Acid-etched surface implants achieved higher resistance to reverse rotation that machined surface implants in a study performed by Klokkevold et al. [15] founding that the dual acid etching conferred a higher resistance (3–4 times greater) to reverse torque rotation compared with the machined surface at intervals of 1, 2, and 3 months in the rabbit femur.

Nowadays, histologic and histomorphometric evaluations are the best methods to evaluate the process of osseointegration, especially the early bone healing phase, and the bone–implant interface [3,11]. Mangano et al. [18] conducted a human histologic study to compare machined and double acid-etched surfaces of implants retrieved of posterior maxilla. After two months of healing, 28 transitional implants (14 dual acid-etched implants and 14 machined implants) were extracted and evaluated. The histomorphometric surface evaluation revealed a higher mean of bone implant-contact (BIC %) in dual acid-etched (37.49 ± 29.51) compared to machined surface implants (21.76 ± 12.79) [18].

Several studies have reported on the clinical outcomes of acid-etched surface implants both in totally and in partially edentulous patients with different types of functional loading [19,20]. A one-year study showed clinical findings of 680 acid-etched implants placed in 212 patients [19]. The implants were subjected to early prosthetic loading (eight weeks in the maxilla and six weeks in the mandible). A total of 298 prostheses were prepared (single crown, fixed bridge, overdentures, and full-arch fixed prosthesis). The survival rate was 98.13%. Implant failures were most frequent in smoker patients (8.6%) [19].

A 7-year retrospective study showed the survival rate of implants subjected to immediate loading or loading within 3 to 10 days after surgery, in totally edentulous patients. A total of 798 acid-etched implants were inserted in 83 patients. The patients were treated consecutively with 4 to 10 implants. 46% of implants were placed in the upper arch and 54% in the lower arch. Sixteen implants (2.1%) were removed because of mobility or infection. Failures were found in 3.2% of maxillary implants and 1.2% of mandibular implants [20].

This clinical study aimed to evaluate the long-term outcomes of the acid-etched surface using a non-submerged surgical procedure and an early loading prosthodontic protocol, intending to propose specific parameters or protocols for a clinical appliance. The clinical relevance of this study is due that not many studies have been published evaluating a high number of implants placed with the same surface, early-loading protocol, and a 17-year follow-up, giving a broad clinical vision of the behavior and predictability expected with this type of treatment.

## 2. Material and Methods

### 2.1. Sample Description

#### 2.1.1. Recruitment

Patients with partial or total edentulism treated at the School of Dentistry, University of Seville, Spain, were considered for the clinical study. This study included patients who required treatment by dental implants. Implants were placed from July 2000 to March 2002.

The study was conducted according to the principles outlined in the Declaration of Helsinki [21] on clinical research involving humans. The ethical committee of the University of Seville approved the study, and all patients signed informed written consent for implant placement. Patients were informed of the clinical protocol, including information related to surgical and prosthetic procedures.

#### 2.1.2. Demographic Description

The study population consisted of 48 patients, 26 males, and 22 females, ranging in age from 30 to 72 years (mean age 52.4 years).

#### 2.1.3. Inclusion and Exclusion Criteria

The inclusion criteria were good systemic health status (ASA I or II) or patients with controlled chronic systemic diseases, a minimum of 8 mm of vertical bone, and a minimum of 7 mm of vestibule-lingual bone (no bone regeneration needed).

The exclusion criteria were the presence of uncontrolled chronic systemic diseases (diabetes, cardiovascular diseases, or others), smoking ≥10cigarettes/day, coagulation disorders, and alcohol or drug abuse.

### 2.2. Diagnosis Records

All treatment diagnoses and planning included diagnostic casts for intermaxillary relations, clinical photographs, panoramic radiographs, and computerized tomography.

### 2.3. Surgery Protocol

One hour before surgery, the patients received prophylactic antibiotic therapy (500 mg amoxicillin and 125 mg clavulanic acid); they also continued taking the antibiotic postoperatively, one dose every eight hours during seven days. After surgery, a chlorhexidine mouthwash was prescribed twice a day for 30 days. Ibuprofen (600 mg), every 12 h, was prescribed for seven days. All patients were treated under local anesthesia using articaine with adrenaline.

A mucosal flap approach was performed. All implants were inserted in a good bone integrity area, and insertion torque was ≥35 Ncm. The implant bed was prepared with standard drills, following the manufacturer’s recommendations (TSA Defcon® screw implants, Impladent, Sentmenat, Spain).

All implants were inserted delayed, in a healed bone with at least six weeks after tooth extraction and with a non-submerged technique. No grafting materials or barriers membranes were used.

After the surgical procedure, all patients received instructions about healing. After implant placement, a healing time of six weeks in mandible and eight weeks in maxilla was developed before prosthetic procedures were started. Early loading was performed when an insertion torque of ≥ 35 Ncm was achieved. Abutments and attachments were inserted, and impressions were made with a silicone material using individual open trays.

### 2.4. Follow-Up

The following patient information was recorded: age, gender, systemic diseases (i.e. cardiovascular disease, diabetes, others), smoking habits (<10 cigarettes/day), periodontitis background, diameter and length of implants, and type of prosthodontic restorations. Follow-up visits were scheduled at three months and six months after prostheses placement and every year during a mean period of 214.4 months (ranged 208–228 months). Marginal bone loss was evaluated based on digital periapical radiographs taken perpendicular to the long axis of the implants.

### 2.5. Implant Features

TSA Defcon® screw implants had a surface treatment with a double acid-etched surface (Figure 1). An internal connection was used for all implants placed.

### 2.6. Success Criteria

The success criteria used for assessment were implant stability and the absence of radiolucency around the implants, mucosal suppuration, and pain.

### 2.7. Statistical Analysis

All available data from all examinations were included in the analyses using SPSS 18.0 (SPSS Inc., Chicago, IL, USA) software. For different parameters, mean values, and standard deviations (SD) were calculated for the descriptive statistics. The chi-square and ANOVA tests were used to compare differences between groups created based on the different risk factors measured. The level of significance was set at 5%.

## 3. Results

One hundred and sixty-nine implants were placed in 48 partially and totally edentulous patients. Of the patients, 22.9% (n = 11) were totally mandibular edentulous and 6.2% (n = 3) totally maxillary edentulous; 11 patients (22.9%) showed a history of periodontal disease; 54.1% of patients (n = 26) were smokers (Table 1). All totally edentulous patients were smokers (chi-square, p = 0,00332). All patients with a history of periodontal disease were smokers (chi-square, p = 0,00087).

Of the 169 implants placed, 99 (58.7%) implants had a diameter of 4 mm, and 70 (41.3%) implants had a diameter of 3,4 mm; 119 (70.4%) implants had a length of 10 mm, and 50 (29.6%) implants were 13 mm in length; 79 (46.7%) implants were inserted in maxilla, and 90 (53.3%) implants were inserted in mandible; 82 (48.5%) implants were inserted in the anterior area and 87 (51.5%) implants were inserted in the posterior area (Table 2).

Sixty-five prostheses were placed in 48 patients over the remaining 164 implants. Thirty single crowns were placed in 15 patients. Twenty-one partially fixed bridges were placed in 18 patients over 2 to 4 implants (51 implants). Five patients (48 implants) were rehabilitated with full-arch fixed prosthesis. 9 patients (35 implants) were rehabilitated with nine overdentures bar, over 3–4 implants. 44 (67.6%) prostheses were screwed, and 21 (32.4%) were cemented. (Table 3) (Figure 2).

During the 17-year follow-up period, twenty-two patients (48.8%) showed complications. We had five early failures with the loss of 5 (2.96%) implants in three patients during the initial healing process, before loading. Seven implants were lost in five patients due to the presence of peri-implantitis during the follow-up period. They were deemed delayed failures, adding a total of 12 (7.1%) lost of the 169 implants, placed in 8 patients, which were lost or had to be removed. The cumulative survival rate for all implants was 92.9%. Implant failures were most frequent in patients with a history of periodontal disease (72.7% vs. 10.8%) and smoker patients (38.4% vs. 9.1%).

In 10 patients (22.8%), 18 implants (10.6%) were associated with peri-implantitis. Peri-implantitis were most frequent in patients with a history of periodontal disease (36.3% vs. 16.2%) and smoker patients (26.9% vs. 13.6%). In 7 implants, the anti-infectious therapy was not successful, and the implant had to be removed.

Twelve patients (26.6%) showed technical complications of prosthetic restorations (fracture of the prosthetic screw, ceramic chipping, resin fracture) over 24 implants (14.2%). Two overdentures and three single crowns had to be repaired. (Table 4).

At the final follow-up, the accumulated mean marginal bone loss was 1.91 mm (SD: 1.24 mm), ranging from 1.1 to 3.6 mm during the time interval from implant insertion to the 17-year follow-up evaluation (Figure 3).

## 4. Discussion

Rehabilitation of partially and totally edentulous patients with prostheses over implants is an established treatment paradigm with predictable outcomes. There is a consensus that prostheses supported by dental implants can achieve a significant improvement in oral function, psychological well-being, and social functioning [21,22].

Implant supported-prostheses in partially and totally edentulous patients has been widely documented in the scientific literature. Some advantages and disadvantages have been attributed to different protocols. The efficacy of these protocols in terms of enhancing the survival of the implants inserted to restore extracted teeth and maintaining a bone and soft tissue stability has been evaluated in recent studies [23,24,25].

This present retrospective study reports on the survival rate of early loaded titanium dental implants with an acid-etched surface in partially and totally edentulous patients. The study yielded an implant survival rate of 92.9%. The present study included the clinical follow-up of 48 patients who received acid-etched surface implants in the School of Dentistry, University of Seville, with a follow-up period of 17 years. Only partially and edentulous patients with no need for bone regeneration were included in this study, so no grafting materials or barriers membranes were used. The clinical protocol included a non-submerged surgical technique and an early loading after 6–8 weeks of healing.

The macroscopic design of implants has a vital role for osseointegration and long-term stability of peri-implant tissues. This aspect may be relevant for improving the primary implant stability when the implant is early loaded [26,27]. The implants placed in this study had a macroscopic design ideal for immediate or early loading. The internal connection was a cone morse type for a one-stage surgical protocol (non-submerged technique) to maintain the crestal bone level [28].

In the present study, after the healing period (6–8 weeks), all patients received abutments that were mounted directly on the implant connection. Clinical outcomes showed that most of the patients (73.3%) were partially edentulous and were treated with single crowns or fixed bridges over 2–4 implants. Among the patients, 26.7% were totally edentulous and were treated with overdentures over 3–4 implants or with full-arch fixed restorations over 9–10 implants. All prostheses were placed with careful attention to design and occlusion. The outcomes of the present study showed a 93.4% cumulative survival rate of prosthetic rehabilitation.

In challenging implant dentistry, such as early and immediate loading, an acceleration of early bone healing might be useful to achieve the osseointegration [18,20,23]. In these clinical situations, acid-etched surface implants have been used with satisfactory survival rates. In a one-year prospective multicenter study, Lazzara et al. [29] inserted 429 double acid-etched implants in 155 patients at 10 study centers. All implants were early loaded (two months) with restorative treatments, including 83 single provisional crowns and 129 fixed bridges supported by two, three, or four implants. Seven implants did not integrate, resulting in a 98.5% cumulative implant survival rate at 12.6 months [29].

Sullivan et al. [30] reported similar results in another study that evaluated the clinical success of restorations supported by acid-etched surface implants with early loading. This 5-year follow-up multicenter study with 197 patients treated with 526 acid-etched surface implants placed in maxilla (34.6%) and mandible (65.4%). 23.0% of the implants were placed in anterior areas, while 77.0% were placed in the posterior zone. The implants were loaded after a healing period of about two months. Prosthesis types included 118 single restorations (118 implants), 134 short fixed bridges (327 implants), and 16 long fixed bridges (81 implants). Eleven implants were lost, resulting in a cumulative success rate of 97.9% at five years [30].

Immediate loading of acid-etched implants has become more widely documented for several clinical situations [31,32]. Peñarrocha et al. [31] reported a success implant rate of 100% after one year in nice partially edentulous patients treated with 54 mandibular implants and immediate loading with full-arch restorations. After surgery, transmucosal abutments were placed and loaded with a provisional acrylic resin prosthesis during the healing period for two months. No complications and a high level of satisfaction were reported [31]. Calvo-Guirado et al. [32] showed the clinical and radiologic evaluation of 86 acid-etched surface implants with expanded platform, inserted in fresh extraction sockets in the maxillary arch of 64 patients. All implants were immediately loaded with provisional restorations. After 15 days, definitive restorations were placed. The results of the study showed an implant success of 97.1% and a limited mean crestal bone loss of 1.01 ± 0.22 mm after 10 years of function, indicating the relevance of acid-etched implant surface and the abutment type for immediate loading [32].

Modifications on surface topography and the chemical surface composition by acid-etching appear to influence the early phases of osseointegration [6,10,13,14]. It is generally accepted that a roughened implant surface results in a better bone tissue response than a machined implant surface. In fact, mechanical and/or chemical treatments on the implant surface can promote osteoblastic differentiation and improve bone anchorage [3,4,6,17].

The surface of TSA Defcon implants used in the present study was roughened using an etching procedure with hydrofluoric and nitric acid [33]. An experimental study demonstrated a homogeneous roughness with Ra mean values of 1.25 ± 0.15 µm and removal torque values after six weeks of 79.7 Ncm for 8 mm implants with 8 mm in length, and 115 Ncm for implants with 10 mm in length. After 12 weeks, these values increased to 101.2 Ncm and 139.7 Ncm, respectively [33].

The long-term outcomes of this study indicate successful bone integration to the implants and agree with the reported results of other experimental and clinical studies with acid-etched surface implants [34,35,36,37]. An in vitro experimental study compared the chemical composition and microstructural configuration of dental implants subjected to two different surface treatments: sandblasting with alumina and acid-etching with hydrofluoric and nitric acids [34]. Scanning electron microscope showed an acid-etched surface without contaminations, while that sandblasted surface was contaminated by the presence of residual alumina used in the sandblasting process. Also, treating the implant surface with an acid-etching technique produces an irregular microtopography surface characterized by higher values of roughness parameters characterized by the presence of deeper valleys and higher peaks than sandblasted surfaces. This study suggested that the microtopography caused by acid-etching stimulates the proliferation and activity of endothelial cells immediately after implant surgery and increases the osteoblast anchorage in the early healing of osseointegration [34].

Topographical features of acid-etched implant surfaces could have a positive effect on the strength of osseointegration [6,7,12,13,14]. An experimental in vivo study showed that osteocytes maintained a straight contact with the acid-etched surface without a gap, and demonstrated a high bone to implant contact ratio (BIC) in acid-etched implants inserted in rat tibia [35]. After four weeks of healing, the acid-etched surface showed a significative increase in the BIC (83.4 ± 5.1%) compared with a machined surface (48.3 ± 13.5%) [34].

The good biological response to acid-etched surfaces of these experimental studies is confirmed with clinical studies than provide essential information about the predictability of treatment with prostheses supported by acid-etched surface implants [36,37,38]. A randomized clinical study evaluated the stability of dental implants with different surface treatments during the osseointegration period. Four types of implants (dual acid-etched; dual acid-etched with nanoparticles; sandblasted and acid-etched; and hydrophilic sandblasted and acid-etched) were tested in 19 patients. After 91 days, only implants with dual acid-etched surface showed a significative correlation of the values of torque insertion and implant quotient stability, demonstrating an acceptable primary and secondary stability [36]. A multicenter clinical study reported an increased success rate of acid-etched implants compared with machined implants in patients with different bone qualities. Two similarly screw implants, one dual acid-etched (247 implants) and the other with a machined surface (185 implants), were placed in 97 patients. After a 3-year post-loading period, 36 implants were lost (12 acid-etched and 24 machined-surface), reporting a cumulative success rate of 95% for the dual-acid etched implants and 86.7% for the machined-surface implants, respectively [37]. More recently, a retrospective study showed the survival and success rates of dental implants with an acid-etched surface. Forty-four patients were treated with 183 acid-etched implants. After 8–10 years of function, five implants were lost, and 178 implants survived. The survival rate was associated with clinical success in 155 implants that did not show signs of inflammation or mucositis/peri-implantitis. Thus, the survival rate reported was 97.3%, and the success rate was 84.7% [38].

Marginal bone loss was considered an essential clinical parameter in the present study. Peri-implant bone remodeling is influenced by several factors, such as macroscopic design, microscopic surface features, and surgical technique [39,40,41]. Crestal bone loss can be altered positively when the implant–abutment interface is horizontally repositioned away from the bone. This technique of platform switching can limit the marginal bone loss in acid-etched implants achieving peri-implant bone stability for a long-term period [32,41].

In the present study, the overall crestal bone loss observed after 17 years was 1.91 mm ±1.24 mm, ranging from 1.1 to 3.6 mm. This clinical finding was in agreement with other studies showing that acid-etched implants had an acceptable marginal bone stability [20,28,30,31,32,41]. Generally, the highest marginal bone loss occurred in the first year after loading, with bone levels becoming more stable afterward [28,41]. In a 7-year study with dual acid-etched implants with external connection inserted crestally and loaded immediately, the mean marginal bone loss found was 1.48 mm. However, bone loss increased notably between 3–12 months [20].

Periodontitis background or smoking habit can be an essential risk factor with adverse effects on implant survival rate [42]. The present study found that implant failures and peri-implantitis were most frequent in patients with a history of periodontal disease and smoker patients. Several studies confirm these clinical outcomes with acid-etched implants [19,38]. In fact, a long-term study showed a similar prevalence of peri-implantitis in patients (22.9%) and implants (11%) with the acid-etched surface. Among the 11 patients with peri-implantitis, four were also smokers. These results suggest that periodontitis background and smoking increase the susceptibility for peri-implantitis due to altered host immune response and a more vulnerable environment of peri-implant tissues [38].

## 5. Conclusions

This long-term follow-up clinical study showed that the early loading of acid-etched implants with different types of prosthodontic rehabilitations demonstrates good treatment outcomes concerning implant and prosthetics survival and marginal bone loss.

Treatment with implants with acid-etched surface and with early loading seems to be indicated in the different types of prosthesis evaluated: single crown, fixed bridge, overdenture, and fixed full-arch.

The most common complication is technical prosthetic complications and the second most common complication is the periimplantitis, appearing mostly in patients with a previous history of periodontal disease and smoker patients.

Within the limitations of this study, we conclude that early loading of acid-etched implants constitutes a clinically predictable treatment when strict selection criteria and clinical planning are applied.

Future studies are needed to assess the behavior of other types of surfaces in different restorative situations with an early-load protocol.

## Figures and Tables

**Figure 1 materials-13-01553-f001:**
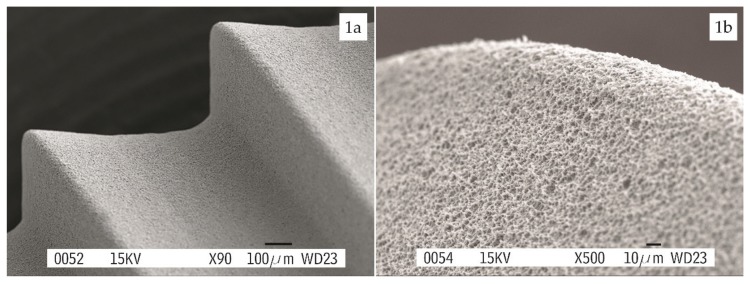
SEM images provided by the manufacturer of the double acid-etched surface topography (TSA Defcon® screw implants, Impladent, Sentmenat, Spain), (**a**) threads macrodesign detail at a magnification of ×90; (**b**) detail of surface on the thread edge at a magnification of ×500.

**Figure 2 materials-13-01553-f002:**
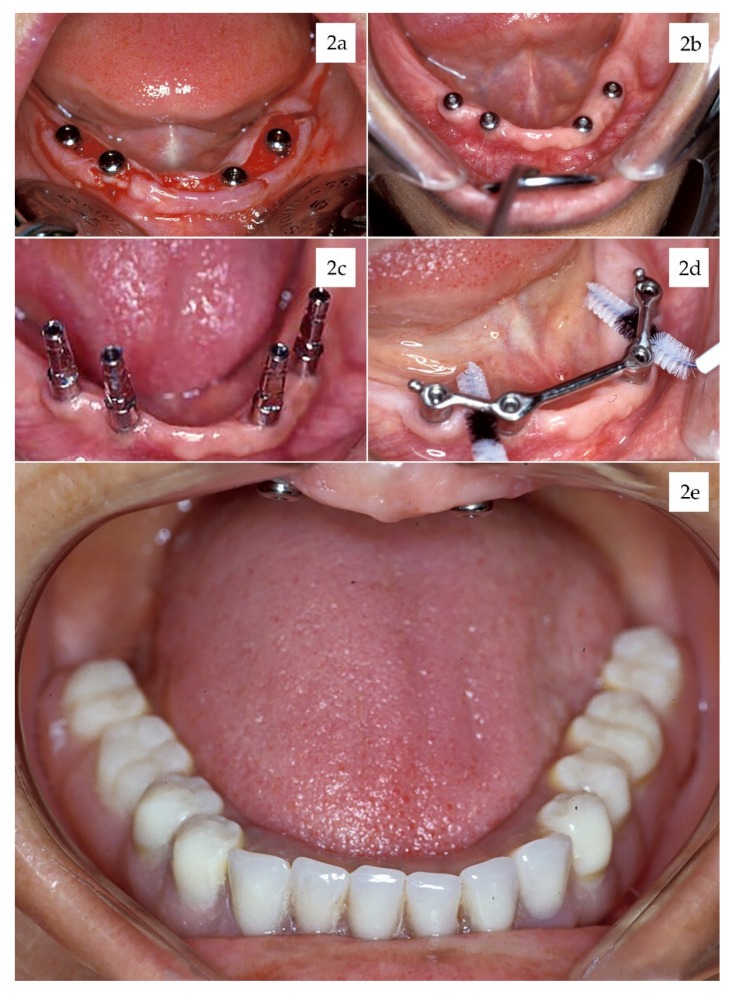
(**a**) Implant placement surgery with mucosal flap approach, (**b**) healing abutments at 6 weeks, (**c**) impression copings, (**d**) cleanable bar, and (**e**) finished overdenture.

**Figure 3 materials-13-01553-f003:**
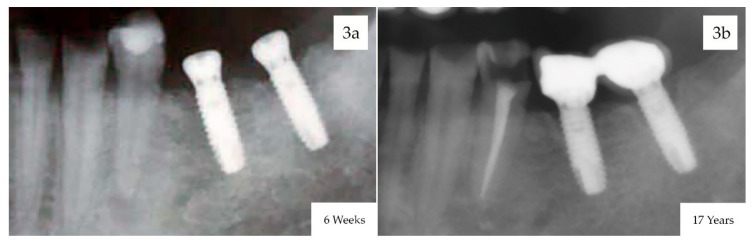
Digital radiographs taken perpendicular to the long axis of the implants used for evaluating marginal bone loss. (**a**) X-rays at 6 weeks after implant placement with the healing abutments ready to start the prosthodontic phase, (**b**) X-rays at 17-years follow-up.

**Table 1 materials-13-01553-t001:** Description of the age and total distribution of the sample, according to the following parameters: gender, smoking habit, periodontitis history, and type of edentulism

Total Patients	n = 48 (100%)	Age	52.4 yrs (Ranged 30–72)
Gender		Male	Female
	n = 48 (100%)	n = 26 (51.4%)	n = 22 (48.6%)
SmokingHabit		Smoker	Nonsmoker
	n = 48 (100%)	n = 26 (51.4%)	n = 26 (51.4%)
Periodontitis History		Yes	No
	n = 48 (100%)	n = 11 (22.9%)	n = 37 (77.1%)
Edentulism		Totally	Partially
	n = 48 (100%)	n = 11 (22.9%)	n = 37 (77.1%)

**Table 2 materials-13-01553-t002:** Description of the total sample implant characteristics: diameter, length, location, area, and percentage of failure and success achieved

Implant Characteristics	N = 169 (100%)	
Diameter	4 mm	3.4 mm
	99 implants (58.7%)	70 implants (41.3%)
Length	13 mm	10 mm
	50 implants (29.6%)	119 implants (70.4%)
Location	Maxilla	Mandible
	79 implants (46.7%)	90 implants (53.3%)
Area	Anterior	Posterior
	82 implants (48.5%)	87 implants (51.5%)
Percentage of Failure/Success	Failure	Success
	12 implants (7.1%)	157 implants (92.9%)

**Table 3 materials-13-01553-t003:** Description of the prosthesis type distribution with the number of implants used to support them and the percentage of screwed and cemented prosthesis.

Number of Implants	Single Crown	Fixed Bridge	Overdenture	Fixed Full-Arch
1	30 (46.1%)	--	--	--
2	--	13 (20.0%)	--	--
3	--	7 (10.7%)	1 (1.5%)	--
4	--	1 (1.5%)	8 (12.3%)	--
9	--	--	--	2 (3.1%)
10	--	--	--	3 (4.6%)
**Total Prostheses**				
65 (100%)	30 (46.1%)	21 (32.3%)	9 (13.8%)	5 (87.6%)
**Screwed /Cemented**				
65 (100%)	44 (67.6%) Screwed		21 (32.4%) Cemented	

**Table 4 materials-13-01553-t004:** Percentage of patients who showed complications during the 17 years follow-up.

Complications	+	-
Patients Showing Complications	22 patients (48.6%)	26 patients (51.4%)
Early Implant Loss	3 patients (6.2%)	45 patients (93.8%)
Delayed Implant Loss	5 patients (10.4%)	43 patients (89.6%)
Peri-Implantitis	10 patients (22.8%)	38 patients (77.2%)
Technical Complications	12 patients (26.6%)	36 patients (73.4%)

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
