# Peer review of "Long-Term Clinical Outcomes of Treatment with Dental Implants with Acid Etched Surface"

_materials, 2020, doi:10.3390/ma13071553_

Round 1

Reviewer 1 Report

Some comments:

  1. The summary should be formulated more precisely and concisely. Key conclusions should be added
  2. Key words: Dental implants →or implant dentistry, use only one and add a long-term evaluation
  3. The aim of the research and what the manuscript wants to inform readers about needs to be improved – more details – see line 99
  4. Improve Fig. 1 into Fig. 1a and Fig. 1b → It is it not clear enough which part in the right SEM image (Fig. 1b) presents the surface area of Fig. 1a. Improve the presentation of the magnification scale – it is not visible enough
  5. Improve Fig. 2 – give numbers for all 5 photos, from 5a to 5e, and write clearly underneath what they represent
  6. Improve Fig. 3 – Fig. 3a and Fig. 3b – with a clear description below (add the magnification scale)
  7. Improve the conclusions - give the most important findings of the study, conclusions, research directions for the future

Author Response

Dear Reviewer,

Thank you for your time and comments that certainly improve the quality and clarity of our work.

Following your recommendations, the following modifications were completed:

1.The summary should be formulated more precisely and concisely. Key conclusions should be added

The abstract has been modified, as suggested.

2. Key words: Dental implants →or implant dentistry, use only one and add a long-term evaluation

Keywords have been changed to Implant dentistry, acid-etched surface, early loading, osseointegration, long-term evaluation, following your recommendation.

3. The aim of the research and what the manuscript wants to inform readers about needs to be improved – more details – see line 99

The objective and clinical relevance have been enhanced by adding more information. The final text is:

“This clinical study aimed to evaluate the long-term outcomes of the acid-etched surface using a non-submerged surgical procedure and an early loading prosthodontic protocol, intending to propose specific parameters or protocols for a clinical appliance. The clinical relevance of this study is due that not many studies have been published evaluating a high number of implants placed with the same surface, early-loading protocol, and a 17-years follow-up, giving a broad clinical vision of the behavior and predictability expected with this type of treatment.”

4.Improve Fig. 1 into Fig. 1a and Fig. 1b → It is it not clear enough which part in the right SEM image (Fig. 1b) presents the surface area of Fig. 1a. Improve the presentation of the magnification scale – it is not visible enough

5.Improve Fig. 2 – give numbers for all 5 photos, from 5a to 5e, and write clearly underneath what they represent

6.Improve Fig. 3 – Fig. 3a and Fig. 3b – with a clear description below (add the magnification scale)

Changes in figures have been made as suggested.

7.Improve the conclusions - give the most important findings of the study, conclusions, research directions for the future

Conclusions have been improved adding the essential findings and future investigation needs

Reviewer 2 Report

Here are my comments and suggestions:

1) Delete the full stop at the end of the Title;

2) No information of the SEM methodology (model, manufacturer, and etc.) as SEM micrographs were presented in Figure 1;

3) English could be improved. For example, Line 49, "the area of contact" could be replaced as "the contact area"; 

Author Response

Dear Reviewer,

Thank you for your time and comments that certainly improve the quality and clarity of our work.

Following your recommendations, the following modifications were completed:

  • Delete the full stop at the end of the Title;

Thank you for reviewing these finest details.

  • No information of the SEM methodology (model, manufacturer, and etc.) as SEM micrographs were presented in Figure 1;

SEM images were provided by the manufacturer as we did not evaluate any parameter regarding surface roughness or any other surface feature, so we don’t have this information. Our study consisted of long-term clinical evaluation.

The text “provided by the manufacturer” has been added to the Figure 1 legend.

  • English could be improved. For example, Line 49, "the area of contact" could be replaced as "the contact area"; 

Changes have been adequately made, as you suggested, and an overall review of the language has been done.
